# The Role of Bypass Surgery for the Management of Complex Intracranial Aneurysms in the Anterior Circulation in the Flow-Diverter Era: A Single-Center Series

**DOI:** 10.3390/brainsci12101339

**Published:** 2022-10-03

**Authors:** Francesco Acerbi, Elio Mazzapicchi, Jacopo Falco, Ignazio Gaspare Vetrano, Francesco Restelli, Giuseppe Faragò, Emanuele La Corte, Giulio Bonomo, Anna Bersano, Isabella Canavero, Marco Gemma, Morgan Broggi, Marco Schiariti, Vanessa Ziliani, Gabriella Raccuia, Salvatore Mangiafico, Giuseppe Ganci, Elisa Ciceri, Paolo Ferroli

**Affiliations:** 1Neurosurgical Unit II, Department of Neurosurgery, Fondazione IRCCS Istituto Neurologico Carlo Besta, 20133 Milan, Italy; 2Experimental Microsurgical Laboratory, Department of Neurosurgery, Fondazione IRCCS Istituto Neurologico Carlo Besta, 20133 Milan, Italy; 3Diagnostic Radiology and Interventional Neuroradiology Unit, Department of Neurosurgery, Fondazione IRCCS Istituto Neurologico Carlo Besta, 20133 Milan, Italy; 4Cerebrovascular Neurological Unit, Department of Neurology, Fondazione IRCCS Istituto Neurologico Carlo Besta, 20133 Milan, Italy; 5Intensive Care Unit, Department of Neurosurgery, Fondazione IRCCS Istituto Neurologico Carlo Besta, 20133 Milan, Italy; 6IRCCS Neuromed, 86077 Pozzilli, Italy

**Keywords:** complex intracranial aneurysms, bypass, ICG videoangiography, combined treatment, CEUS, fluorescein

## Abstract

Despite the increasing popularity of flow diverters (FDs) as an endovascular option for intracranial aneurysms, the treatment of complex aneurysms still represents a challenge. Combined strategies using a flow-preservation bypass could be considered in selected cases. In this study, we retrospectively reviewed our series of patients with complex intracranial aneurysms submitted to bypass. From January 2015 to May 2022, 23 patients were selected. We identified 11 cases (47.8%) of MCA, 6 cases (26.1%) of ACA and 6 cases (26.1%) of ICA aneurysms. The mean maximal diameter was 22.73 ± 12.16 mm, 8 were considered as giant, 9 were fusiform, 8 presented intraluminal thrombosis, 10 presented wall calcification, and 18 involved major branches or perforating arteries. Twenty-five bypass procedures were performed in 23 patients (two EC–IC bypasses with radial artery graft, seventeen single- or double-barrel STA–MCA bypasses and six IC–IC bypasses in anterior cerebral arteries). The long-term bypass patency rate was 94.5%, and the total aneurysm exclusion was 95.6%, with a mean follow-up of 28 months. Median KPS values at last follow-up was 90, and a favorable outcome (KPS ≥ 70 and mRS ≤ 2) was obtained in 87% of the cases. The use of bypass techniques represents, in selected cases, a valid therapeutic option in the management of complex anterior circulation aneurysms when a simpler direct approach, including the use of FD, is considered not feasible.

## 1. Introduction

The correct management and treatment of unruptured intracranial aneurysms (UIAs) is a topic of great importance. The widespread diffusion of neuroimaging modalities has led to an increased prevalence of UIAs diagnosis from 1–2% to 8.8% [1,2]. Although prediction models based on different risk factors estimated a 3- to 5-year risk of aneurysm rupture ranging from >1% to 15–20%, a precise estimation of lifelong UIAs rupture risk for each patient is still not feasible [3,4,5]. Thus, decision making regarding the best managing options for patients with UIAs is very difficult. In the very recent European Stroke Organization (ESO) guidelines on management of unruptured intracranial aneurysms, it has been suggested that, in adult patients in whom the estimated 5 years risk of aneurysm rupture is higher than the risk of preventive treatment, the latter should be proposed, considering the most effective and safe modality for that particular aneurysm [6]. In the same guidelines, the authors suggested considering aneurysm morphology and complexity as important factors when balancing between conservative and preventive treatment [6]. In fact, despite advancement in surgical and endovascular techniques, the treatment of complex intracranial aneurysms still poses significant difficulties for vascular neurosurgeons and interventional neuroradiologists [7,8,9,10]. Aneurysm complexity depends on several factors, such as size, shape, presence of intraluminal thrombus, wall calcification or atherosclerosis and involvement of major branches or perforators in the aneurysm sac. When dealing with such aneurysms, a direct endovascular or surgical occlusion may not be feasible [11]. Although flow diverters (FDs) have gained popularity as an endovascular treatment option for intracranial aneurysms, when dealing with more complex aneurysms the results are uncertain, and there is a higher rate of complications [12]. In these cases, different approaches may be considered. In particular, a bypass strategy could be used to protect one or more cerebral territories distal to the aneurysm that can be secondarily completely or partially occluded through a surgical or endovascular approach [13,14,15,16,17,18,19]. This strategy should be tailored to the specific patient’s anatomy and aneurysm characteristics [20]. Only a few case series of complex intracranial aneurysms treated using bypass surgery have been reported in the literature [14,15,19,21,22,23], and very limited cases have specifically addressed the impact of FD stenting on the indication of bypass surgery in unruptured anterior circulation aneurysms. Therefore, we retrospectively reviewed our case series of consecutive patients with complex intracranial aneurysms in the anterior circulation, submitted to bypass at our institution in the period 2015–2022, well after the introduction of FD, to analyze their specific indications, different treatment strategies, technical nuances, complication rate and long-term follow-up.

## 2. Materials and Methods

### 2.1. Patient Selection

We retrospectively reviewed the vascular section of the surgical database of the Neurosurgical Department of the Fondazione IRCCS Istituto Neurologico Carlo Besta in the period January 2015–May 2022, to find all cases of complex intracranial aneurysms in which a cerebral bypass was performed as a part of the strategy to manage the case. We defined a complex intracranial aneurysm based on the following criteria: (a) giant aneurysms, i.e., with a diameter larger than 25 mm; (b) fusiform or serpentine aneurysms; (c) aneurysms with intraluminal thrombosis and/or wall calcification; (d) major branch or perforating arteries arising from the aneurysm sac.

Each of these cases was preoperatively reviewed by our multidisciplinary neurovascular team including two expert neurovascular surgeons (FA and PF) and interventional neuroradiologists (GF, SM, GG, EC), who decided that a multimodal strategy including surgical revascularization would carry the best chance for a favorable outcome.

The Ethical Committee of the Foundation IRCCS Istituto Neurologico Carlo Besta approved the prospective surgical database of the Neurosurgical Department (4 April 2012) that was used to perform this retrospective analysis. Three cases from this series have already been described in three separate papers, addressing specific technical issues of each case [20,24,25].

### 2.2. Bypass Strategy

The bypass modality was tailored to the aneurysm characteristics and the brain territory to be supplied (Table 1).

For aneurysms located in the middle cerebral artery (MCA), a superior temporal artery (STA)-MCA single- or double-barrel bypass was considered, depending on the need to supply one major branch or all MCA territory. In addition, intracranial–intracranial (IC–IC) bypass, taking advantage of the close proximity of other MCA branches not directly coming from the aneurysm, was also hypothesized.

For aneurysms located in the internal carotid artery (ICA), we chose the strategy based on the results of balloon test occlusion (BTO) with hypotensive challenge and/or patient’s specific anatomy (i.e., the presence and function of anterior (AcommA) and posterior (PcommA) communicating artery), considering also the preoperative flow measurement evaluation by MR-angio NOVA (Philips/3D-ACHIEVA DSTREAM; Amsterdam, Netherlands). Specifically, when patients presented a neurological deficit only after hypotensive challenge and only the MCA territory needed to be protected, a double-barrel superior temporal artery (STA)-MCA bypass was considered. When the patient clinically failed the basal BTO and both MCA and ACA territory had to be protected, a high-flow extracranial (EC)-IC bypass with radial artery graft between external carotid and M2/M1 segments was planned. Finally, in cases in which the aneurysm was associated with the presence of local cerebral hypoperfusion, a simple STA–MCA or occipital artery (OA)-MCA bypass was also considered as an option.

In cases of aneurysms of anterior communicating artery (AcommA) or distal anterior cerebral artery (ACA), in situ side-to-side PerA–PerA bypass alone or combined with side-to-side callosomarginal artery (CmaA)–CmaA bypass or interpositional bypass with STA graft to the CmaA was used, based on which arteries needed to be replaced [20,24].

All patients were operated on under antiplatelets therapy (Acetylsalicylic acid 100 mg), either already taken as chronic therapy or started before the operation from 7 days to 1 day preoperatively. Surgery was performed under general total intravenous anesthesia with propofol and remifentanil, normothermia and maintenance of normal values of blood pressure, with PaCO^2^ between 35 and 40 mmHg. Intraoperative electrophysiological monitoring was always performed. During temporary clipping, systolic blood pressure was raised 20% above the preclipping value by means of norepinephrine administration, and burst suppression was also reached by increasing propofol dosage.

In all cases, bypass patency and good brain perfusion were intraoperatively assessed by indocyanine green videoangiography and FLOW800 analysis (Pentero 900 or Kinevo microscope, Carl Zeiss Meditec AG, Oberkochen, Germany) [20,25]; in selected cases, fluorescein videoangiography (Pentero 900 or Kinevo microscope, Carl Zeiss Meditec AG, Oberkochen, Germany), contrast-enhanced US (CeUS) (MyLab [Esaote, Genoa, Italy]) [25], quantitative Charbel Flow probe (Transonic, Ithaca, NY, USA) and neuronavigation (Stealth S7 or S8 Medtronic Inc. Minneapolis, MN, USA) were also performed.

### 2.3. Aneurysm Occlusion

The modality of aneurysm occlusion was related to its characteristics and to the type of artery involved (Table 1).

In particular, aneurysm’s occlusion was surgically planned by clip reconstruction or complete or partial trapping [13] in all cases of complex MCA aneurysm and for selected ICA aneurysms, in the same surgical session as the bypass.

For all other cases of ICA and ACA aneurysms, endovascular treatment (see below) was instead considered in a separate session, after bypass patency and good distal territory supply was confirmed by postoperative digital subtraction angiography (DSA), usually the day after the surgical treatment.

### 2.4. Clinical and Radiological Pre- and Postoperative Management

All patients were submitted to pre- and postoperative clinical evaluation including neurological examination, following normal clinical practice and based on our protocol for management of surgical patients, as previously described [26,27,28]. In particular, the clinical status of the patients was evaluated using the modified Rankin Scale (mRS) and the Karnofsky Performance Scale (KPS) at admission and at the last follow-up period [26,27,28]. We defined a favorable clinical outcome as KPS ≥ 70 and mRS ≤ 2.

Preoperative radiological management included DSA (SIEMENS/Multiplanar, Munich, Germany) with 3D reconstruction to evaluate the characteristics of the aneurysm and the surrounding branches, CT-angio (GE Revolution/Dual Energy, GE, Boston, MA, USA) to better highlight the relationship between the aneurysm and the brain and MRI or MR-angio with NOVA (Philips/3D-ACHIEVA DSTREAM) to better evaluate the aneurysmal features, to exclude the presence of ischemic lesions and to measure baseline flow in all main arteries of the polygon of Willis. BTO was performed when the aneurysm was located on the ICA, as above described.

Postoperative radiological assessment included immediate postoperative CT-angio, to confirm immediate postoperative bypass patency and rule out possible operative complications, and DSA to further confirm bypass patency and dynamic revascularization assured by bypass. DSA was also part of the endovascular treatment to manage the aneurysms in cases of combined treatment. In selected cases, MR with MR NOVA was performed to rule out ischemic complication and quantitatively assess the blood flow in the bypass and in other vessels of the circle of Willis.

During late follow-up after treatment, patients were regularly clinically assessed and submitted to periodic CT-angio or MR-angio with NOVA.

### 2.5. Statistical Analysis

Continuous variables are presented as the mean ± standard deviation. Binary variables are expressed in %.

## 3. Results

### 3.1. Patients Characteristics

From January 2015 to May 2022, 255 patients submitted to neurosurgical or endovascular occlusion of intracranial aneurysms at the Department of Neurosurgery of the Foundation IRCCS Istituto Neurologico Carlo Besta.

Of these, 23 patients presented with complex intracranial aneurysms in the anterior circulation, where a bypass was chosen as part of the therapeutic strategy to occlude the aneurysm while preserving distal perfusion (Table 1).

MCA aneurysms were more common (11 cases, 47.8%), followed by ACA (6 cases, 26.1%) and ICA (6 cases, 26.1%). There were nine cases in the left hemisphere, thirteen in the right and one case in the middle (AcommA). The mean maximal diameter of the aneurysms was 22.73 ± 12.16 mm. Eight aneurysms were considered as giant, nine were fusiform, eight presented intraluminal thrombosis, and ten presented with wall calcification. Finally, in 18 cases, there were major branches or perforating arteries arising from the aneurysm sac. Four patients (patients no. 5, 9, 10 and 15) were previously treated at other hospitals, and presented to our attention with recanalization after initial complete obliteration with stent-assisted coil embolization (patients 5, 10 and 15) or after partial coiling (patient no. 9).

Preoperative clinical characteristics are summarized in Table 2.

### 3.2. Bypass Strategies

Twenty-four bypass procedures were performed in 23 patients (Table 1), as follows:A.In two patients, an EC–IC bypass with radial artery graft was performed, connecting one of the branches of the external carotid artery (ECA) with the M1 segment of the MCA (patients no. 4 and 16). Both these patients harbored aneurysms in the ICA, of 18 and 25 mm, respectively. Patient no. 4 presented a right large cavernous ICA aneurysm, with symptoms related to mass effect. After she failed the BTO with the need to replace flow in both ACA and MCA territory, a revascularization strategy with high-flow EC–IC bypass with radial artery graft was performed, with subsequent surgical trapping of the aneurysm (the patient presented a stage IV renal insufficiency, preventing an endovascular approach from being safely performed). Patient no. 16 presented with two giant bilateral aneurysms of the cavernous ICA, with mass effect and symptoms related to the left one. Before coming to our attention, after an episode of significant headache, a spontaneous right ICA thrombosis with aneurysm occlusion was found. Therefore, due to the need to replace the only artery assuring flow in the anterior circulation, a high-flow left EC–IC bypass with radial artery graft was performed, with subsequent endovascular occlusion of the left ICA, including the cavernous aneurysm.B.In eight patients, a double-barrel STA–MCA bypass was performed. In most of these cases (patients no. 1, 3, 8, 9, 12 and 14), this type of bypass was chosen for a complex MCA aneurysm that involvedtwo large branches of the MCA and whose flow needed to be preserved before addressing the aneurysm sac. In two cases (patients no. 7 and 13), this type of bypass was instead performed in complex ICA aneurysms where the BTO demonstrated a neurological deficit only after hypotensive challenge (see above).C.In six patients, a simple STA–MCA bypass was performed. In two cases (patients no. 2 and 17), this type of bypass was performed to replace a single MCA branch involved in the aneurysm sac as part of the occlusion strategy (Figure 1). In two cases (patients no. 19 and 20), STA–MCA bypasses were performed after having addressed a partially thrombosed aneurysm of the MCA with a temporary clipping, removal of the thrombus and clip reconstruction, with ICG videoangiographic evidence of occlusion of an MCA branch coming out from the aneurysm sac; in these cases, the parietal branch of the STA had been prepared at the beginning of surgery to be ready in case a bypass was needed. In one case (patient no. 6), the patient presented a giant, partially thrombosed and calcified aneurysm of the left ICA, compressing the M1 tract of the left MCA, with a resultant hypoperfusion in the MCA territory and recurrent transient ischemic attacks (TIAs); the STA–MCA bypass was therefore performed to correct the hypoperfusion in the MCA territory before endovascular occlusion of the ICA aneurysm (Figure 2). In one case (patient no. 22), the patient presented a giant aneurysm of the supraclinoid prebifurcation ICA, and the STA–MCA bypass was used to replace flow in the MCA territory before partial trapping of the aneurysm, taking into consideration that flow in the ACA territory was assured by a large AcommA (Figure 3).

D.In one case (patient no. 21), a combined bypass strategy was performed for a complex MCA aneurysm, in which an STA–MCA bypass was used to preserve flow in one of the branches coming out from the MCA aneurysm, while the second larger branch was re-connected (IC–IC bypass) with the proximal afferent artery though an end-to-end microanastomosis after sectioning of the unclippable aneurysm sac from the MCA circulation.E.In three patients (patients no. 10, 15 and 18), a side-to-side pericallosal artery–pericallosal artery (perA–perA) bypass was performed to preserve distal flow in the ACA territory, before endovascular treatment of complex aneurysm in the proximal ACA (Figure 4).

F.In three patients, a combined procedure involving multiple bypasses to preserve flow in different territories of the distal ACA was performed. In one case (patient no. 5), a side-to-side perA–perA bypass was performed together with a right STA–CmaA artery bypass using a contralateral STA as a graft, to replace both perA and CmaA territories before endovascularly occluding a complex aneurysm of the proximal right ACA that was already submitted to an unsuccessful endovascular treatment in another institution [24]. In one case (patient no. 11), PerA–PerA and CmaA–CmaA artery bypasses were performed in the same patient to preserve the two distal territories in a complex large aneurysm of the proximal ACA, involving both pericallosal and callosomarginal arteries [20]. In one case (patient no. 23), a similar strategy was planned. However, after intraoperative thrombosis of perA–perA bypass occurred two times, a salvage strategy was considered, by grafting the STA segment at its bifurcation into parietal and frontal branches to create a bridge between the right perA proximal to the thrombus and the two perA distal to the thrombus.G.In one case, an OA–MCA was performed in the same patient (no. 16) that previously received a high-flow EC–IC bypass with radial artery graft, due to the reduced flow in the graft for subsequent vasospasm, with left hemispheric hypoperfusion.

### 3.3. Strategy of Aneurysm Occlusion

The actual strategy of aneurysms’ occlusion was also tailored to the location and shape of the aneurysm, as well as the involvement of perforators (Table 1).

#### 3.3.1. MCA Aneurysms

Whenever possible, a complete surgical trapping in the same surgical procedure, after bypass completion, was considered as the first choice for complex MCA aneurysms not involving lenticulostriate arteries. This was performed on six MCA aneurysms located distally to lenticulostriate perforators (patients no. 8, 9, 12, 17, 19 and 21). In one of these cases (patient no. 19), a complete trapping was performed only after an unsuccessful first attempt of aneurysm clipping had been already performed, with subsequent need of STA–MCA bypass. In one case (patient no. 2), this initial strategy was intraoperatively modified due to the extreme adherence between the aneurysm sac and the functional brain parenchyma (left temporal and parietal lobes), which made a visualization of the proximal arterial segment afferent to the aneurysm sac not safe. For this reason, the strategy switched to clipping of the efferent artery only, immediately distal to the aneurysm (partial distal trapping), controlling the effect on aneurysm flow by intraoperative CeUS and Doppler, which showed no residual flow already in the OR [25]. In another case (patient no. 20), a complete trapping was not considered necessary, as the aneurysm could be clipped on its neck, with sacrifice of one temporal branch that was replaced by distal STA–MCA bypass.

In three cases of complex MCA aneurysms involving perforators, the initial strategy was to perform only a partial distal trapping after bypass completion, in order to manipulate flow and induce intra-aneurysms progressive thrombosis, while possibly maintaining flow in the perforators [29,30,31,32,33,34,35,36]. In one case (patient no. 3), this strategy was successfully completed. In two other cases (patients no. 1 and 14), we had to perform a complete trapping due to the intraoperative rupture of the aneurysms sac during surgical manipulation after having already performed the distal bypass.

#### 3.3.2. ACA Aneurysms

For distal ACA aneurysms, involving the origin of CmaA and PerA (patients no. 5, 11 and 23), our first choice was to perform endovascular occlusion of the aneurysm together with the parent distal ACA, in a separate surgical procedure, one day after the bypass. This type of occlusion was effectively performed in patients no. 5 and 11 [20]. In patient no. 23, due to intraoperative thrombosis of the bypasses, a complete immediate aneurysm occlusion could not be performed. This patient, after having completely recovered from the initial ischemic complication related to PerA occlusion, was submitted to an endovascular FD positioning (silk vista baby), four months after the bypass.

For proximal ACA aneurysms involving AcommA complex (patients 10, 15, 18), an endovascular occlusion of the aneurysm sac together with dominant A1 was performed, one or a few days after the bypass (Figure 4).

#### 3.3.3. ICA Aneurysms

In three cases (patients no. 7, 13 and 16), aneurysms occlusion was performed together with the ICA by endovascular means, in a separate surgical procedure, one day after the bypass. In one case (patient no. 6), as a STA–MCA bypass was performed due to hypoperfusion in the MCA territory from an MCA stenosis directly related to the aneurysm’s mass effect (see above), the left giant ICA aneurysm was endovascularly addressed a few weeks after the surgical procedure, by positioning a FD (Derivo). In one case (patient no. 4), a complete surgical trapping was performed by proximal clipping of the ICA at the neck and distal clipping on a supraclinoid, pre-ophtalmic, ICA segment. In the last ICA case (patient no. 22), the giant aneurysm could not be completely trapped, as the posterior communicating and anterior choroidal arteries were originating from the posterior part of the aneurysm sac. Therefore, a partial trapping was surgically completed in the same surgical procedure, after the bypass, by positioning of a proximal clip on the supraclinoid post-ophthalmic ICA segment, and two distal clips, one on A1 and one on M1, to maintain flow from the posterior communicating to anterior choroidal artery (Figure 3).

### 3.4. Periprocedural Complications

Intraoperative complications occurred in four patients. In two complex MCA (patients no. 1 and no. 14), as above described, the aneurysm sac ruptured during surgical manipulation, performed after bypass completion (one of these aneurysms, in patient no. 14, had previously bled). This caused a change in surgical strategy, with the need to perform a complete trapping of the aneurysm sac. In patient no. 1, an internal capsule stroke therefore occurred, as a consequence of the occlusion of lenticulostriate arteries. In patient no. 14, too, an internal capsule stroke was evident as a consequence of the occlusion of lenticulostriate arteries; however, this patient unfortunately suffered from a bilateral hemorrhagic lesion in the brainstem, determining a significant worsening of her neurological condition, characterized by deep coma. In one patient (no. 9), an extradural hematoma was found at the final phase of surgery, posteriorly to the pterional craniotomy, needing an enlargement of bone exposure and hematoma evacuation, without clinical consequences. In patient no. 23, the PerA–PerA bypass intraoperatively thrombosed and remains occluded although it was washed with intraarterial heparinized solution and performed two times; therefore, the CMA–CMA bypass was not completed, and another surgical strategy was pursued by means of using a STA (both branches) graft between proximal ACA and distal territories: unfortunately, these vessels also faced intraluminal occlusion. No intraoperative neurological monitoring variations were detected; the surgical procedure was interrupted: a postoperative course was characterized by worsening of neurological status due to frontal lobe ischemia, with subsequent late clinical improvement. A thrombophilic screening was performed and resulted negative; however, the patient was operated on shortly after symptomatic COVID-19 infection, and this was the only factor possibly associated with a hyperthrombotic status [37]. Four months later, a FD was positioned as a curative attempt.

The postoperative course was uneventful in most of the other patients. In two cases, an internal capsule stroke occurred: in patient no. 3, the pathogenesis was related to the thrombosis of a single lenticulostriate artery originating from the fusiform aneurysmal sac, after the distal clipping. In patient no. 4, the stroke was probably a consequence of microembolic phenomena during surgical manipulation, since no perforators occlusion was detected at early DSA. Patient no. 16 presented radial artery graft vasospasm in IX POD; the case was managed with intra-arterial nimodipine administration and several angioplasties. As a long-term consequence of this event, the arterial graft presented a reduction in the blood support: the patient suffered multiple TIAs from hypoperfusion in the left hemisphere, and therefore, a left OA–MCA bypass was performed. In patient no. 18, after endovascular treatment, a right cerebellar hemorrhage occurred with IV ventricle compression and obstructive hydrocephalus: the patient was scheduled for urgent EVD, which was subsequently converted to a VP shunt; no surgical treatment of the hematoma was needed.

### 3.5. Bypass Patency, Short- and Long-Term Results

Mean duration of follow-up was 28 months (range 3–84 months).

Immediate intraoperative bypass patency was confirmed by ICG, fluorescein videoangiography, and CeUS in 22 out of 23 cases (95.6%). Postoperative CT-angio and DSA confirmed patency of the bypass on the first postoperative day in 22 out of the 22 cases with intraoperative patency (100% of the cases). Long-term patency was evident in 21 out the 24 bypasses (87.4%) and 21 of 22 cases (95.4%, considering that in patient no. 5, one of the two bypass was occluded at the last F-U, while in patient no. 16, both bypasses were occluded).

Complete exclusion of the aneurysm from brain circulation was obtained in 22 out of 23 cases at last F-U (95.6%).

Median KPS at hospital discharge was 90. At long-term F-U, a favorable clinical outcome (KPS ≥ 70 and mRS ≤ 2) was obtained in 20 out of 23 cases (Table 3). Two patients were dead at long-term F-U: one (patient no. 4) died 3 months after surgery from an unrelated cause (severe renal insufficiency); one (patient no. 1) died as a consequence of a brainstem hemorrhage 6 months after surgery. One patient (no. 3) still presented at 3 years post-op with a KPS of 40 m and mRS of 4 due to both the consequences of a right hemispheric peri-operative stroke and progressive cerebral atrophy, considered as a possible expression of fronto-temporal dementia.

## 4. Discussion

We described a series of patients with complex aneurysms in the anterior circulation, where a bypass was performed as a fundamental part of the treatment strategy in order to preserve distal flow before addressing the aneurysm, when it was considered not suitable for direct surgical or endovascular technique. We showed different bypass techniques, demonstrating a high rate of immediate and long-term bypass patency and a low rate of peri- and postoperative complications. The use of bypass to preserve distal outflow allowed for addressing anterior circulation aneurysms with different treatment modalities, based on the location and involvement of perforators, using both surgical and endovascular techniques.

We defined a complex aneurysm based on its size being larger than 25 mm, its shape (fusiform or serpentine), the presence of intraluminal thrombosis or wall calcification, and a major branch or perforating arteries arising from the aneurysm sac. Indeed, the mean maximal diameter of the aneurysms in our series was 22.73 ± 12.16 mm, and most of the cases (78.2%) presented major branches or perforators arising from the aneurysm sac. In addition, four cases had been previously submitted to unsuccessful endovascular attempts, which further increased aneurysm complexity. Although there is no uniformity in the definition of aneurysm complexity in the literature [13], the characteristics defined in our series made these cases particularly challenging for a direct microsurgical or endovascular treatment [38]. These aneurysms represented around 9% of the entire series of aneurysms treated during the same period at our institution.

After flow diverters (FD) were demonstrated to represent a safe and efficient solution for the treatment of complex intracranial aneurysms located from the petrous to hypophyseal segments of the ICA [9] by inducing vessel wall remodeling and aneurysm occlusion, its use rapidly spread to other locations [39] and also in some cases to acute settings [40]. However, its use has not completely resolved the problems of aneurysm complexity. In particular, it has been shown that the presence of a large branch vessel arising from the aneurysm sac or a fusiform shape, as in most of our cases of complex aneurysms, represents one of the most important predictors of aneurysm persistence after treatment with FD [41]. In addition, although treatment of aneurysms located distally to ICA, such as on ACA or MCA, has been reported with different degrees of success and complications [8,10,42], these indications are still considered off-label in the USA, and further studies need to be published before any definitive conclusion can be drawn regarding safety and efficacy of FD in these location [41].

An alternative approach to treat these challenging lesions is represented by using different forms of cerebral revascularization to preserve distal flow, associated with parent artery occlusion to provide a definitive treatment for the aneurysm sac [43,44,45,46,47]. In addition, distal flow preservation through bypass techniques could also be used as part of the treatment strategy when a sacrifice of one large arterial branch is needed to completely exclude the aneurysm from brain circulation.

Our retrospective study had the aim of evaluating the indications and the results of the use of different revascularization techniques as part of the multimodality management of patients presenting with complex intracranial aneurysms in the anterior circulation. Our series included in the great majority (69.6%) complex aneurysms of MCA or ACA, where FD still presents some intrinsic limitations that make its use not the first choice for all cases [8,9,10,41,42]. In addition, also in the complex ICA aneurysms, our neurovascular team considered a direct surgical or endovascular approach, including FD, as not ideal to completely exclude the aneurysm from brain circulation, especially where an endovascular approach was considered unsuitable for a stage IV renal insufficiency (patient no. 4). In addition, although the aneurysm could be treated by using a flow-diverting strategy, in one patient (no. 6), the aneurysm was so large and calcified that it compressed the M1 segment of the MCA, causing a left hemispheric symptomatic hypoperfusion. This needed be addressed first with an STA–MCA bypass, before the aneurysm could be successfully treated by our endovascular colleagues with a FD. The idea of treating an aneurysm in a more definite way is clearly very important, particularly in unruptured cases, where the effect of treatment must be balanced against the natural course of the disease [48,49,50,51].

By mastering the three types of vascular microanastomosis (i.e., end-to-end, end-to-side and side-to-side) [52] it is possible to hypothesize different creative forms of revascularization, including EC–IC or IC–IC bypasses, that could be adapted and tailored to the specific patient’s anatomy and to the precise need for flow revascularization [20,24]. When feasible, we preferred avoiding the use of interpositioning grafts, as they have potential disadvantages related to a lower patency rate with a higher risk of failure [53], the need for multiple microanastomosis and the complications of additional procedures for graft harvesting and possible donor sites (at least for the radial artery or the saphenous vein) [52]. On the contrary, we favored a simpler approach through double-barrel or single-barrel STA–MCA bypasses that, although traditionally considered low-flow bypasses [54,55], have the potential to provide higher flow, depending on brain demand [56]. As an alternative, for flow preservation in distal ACA territory, although considered more challenging due to the depth and narrowness of the interhemispheric approach [57], we prefer in situ microanastomosis, due to the higher patency rate compared to the use of arterial or venous graft and because these bypasses are less vulnerable to external impact, being safe from neck torsion, injury and occlusion with external compression [58]. As a matter of fact, a high-flow EC–IC bypass with interpositional radial artery graft was performed in only two of our cases (patients no. 4 and 16), while all the other revascularizations were performed by using simpler STA–MCA or in situ IC–IC bypasses. This strategy was associated with a high rate of intraoperative bypass patency (95.6%), as confirmed by ICG and fluorescein videoangiography, and CeUS, and this was maintained in most of the cases at long-term radiological follow-up (95.4%).

The different revascularization strategies were the pre-requisites to treat the aneurysms afterwards in the most efficient ways. Whenever possible, a complete surgical trapping should be preferred, particularly for complex MCA or ICA aneurysms, because it can be performed in the same surgical approach [31,35,59,60,61,62]. We were able to perform a complete aneurysm trapping after revascularization in seven complex MCA and in one ICA aneurysms. However, partial trapping, i.e., proximal inflow or distal outflow occlusion, to reduce aneurysm’s flow and induce intra-aneurysmal thrombosis, may be considered as a reasonable alternative when cortical branches or perforators arise from the aneurysmal sac [63]. In these cases, we prefer using partial distal occlusion, in order to allow anterograde flow in the branches proximal to the aneurysm and concomitantly induce progressive aneurysmal thrombosis [14,60,64]. An important limitation of this strategy is related to the difficulties in predicting both the extent and the speed of aneurysmal thrombosis and the permanent good blood flow supply to the perforators of the branches that need to remain patent [19,32]. Although we routinely assess flow in arteries and brain parenchyma intraoperatively by means of ICG-VA and FLOW 800 software, this could only give information about the area exposed during the surgical approach, and it cannot substitute for a standardized evaluation of preoperative and postoperative blood flow in the regions vascularized by the bypass [53,65]. We used partial distal trapping in two complex MCA cases, in which the aneurysm was embedded in eloquent tissue, preventing a safe exposure of its proximal end (patient no. 2), or perforators were involved in the aneurysm sac (patient no. 3). In the first case, we obtained immediate intraoperative thrombosis of the aneurysm sac that was demonstrated by using intraoperative CeUS with Doppler and subsequently confirmed by postoperative DSA and MRI [25]. In the second case, a capsular stroke due to the occlusion of one lenticulostriate artery originating from the aneurysm sac occurred 24 h after surgery, during the progressive thrombotic changes inside the fusiform aneurysm sac. In addition, a partial trapping was also performed in one giant ICA aneurysm (patient no. 22) by clipping the main inflow (ICA) and the two main outflows (M1 and A1), while leaving a patent posterior communicating artery to provide flow in the anterior choroidal artery, both originating from the aneurysm sac: in this case, the aneurysm underwent a progressive shrinkage, while posterior communicating and anterior choroidal arteries remained patent at long-term F-U. On the contrary, most of the complex ICA and all ACA aneurysms were occluded by endovascular means, in a separate surgical session after the revascularization procedures. This strategy was chosen for ICA aneurysms, in which proximal exposure was considered more difficult to obtain by a direct surgical approach. In addition, this was also the preferred choice for anterior communicating or anterior cerebral arteries. In fact, although it is possible to perform both the bypass and the surgical trapping in a single session [63], in more proximal aneurysms, this would require either a large bifrontal exposure or two distinct craniotomies [66]. Thus, in order to avoid excessive vessels manipulation in a single session, we felt that in these cases a combined surgical and endovascular procedure could be considered safer.

In our experience, the combination of multiple intraoperative visualization tools is of paramount importance during complex vascular cases, which require multimodal management; in particular, each tool can provide specific information, complementary to each other, to better understand flow dynamics and vessels anatomy during surgery. We found that CEUS allowed for studying the characteristics of the aneurysm, its calcification and the vascular anatomy even at distant sites not directly exposed by the surgical approach [20,25]. In addition, ICG videoangiography with FLOW 800 analysis and SF videoangiography [20,25] allowed for intraoperatively evaluating bypass patency and the absence of parenchymal hypoperfusion. They could be performed in a synergistic way, because FLOW 800 software provided by Zeiss has the peculiarity of allowing semiquantitative flow measurement [67], while the definition of small perforators and tiny cortical vessel is much higher with SF-VA [29,68]. Our group showed the possible combination of both tracers on a large cohort of patients, without any adverse drug effect [69,70,71].

### Limitations and Future Perspective

This study has several limitations. First of all, it is a retrospective case series, with an intrinsic selection bias related to the fact that we decided to consider only complex aneurysms not amenable to simple microsurgical or endovascular approach, including the use of FD. Therefore, no control group with different strategies in similar aneurysms could be used as a comparison. Furthermore, the relatively low number of cases and the heterogeneity of the series, together with the fact that a good outcome was obtained at last follow-up in most of the patients, prevented us from performing a subgroup analysis evaluating possible predictors of poor outcome, including hemodynamics and geometry of the bypass. Finally, patients were followed up with for a relatively short period of time, and no definitive information about very long-term bypass patency could be retrieved.

In the future, computational fluid dynamics, which has been recently applied to more accurately predict the risk of rupture of UIAs [72,73], could also be used more extensively to model complex intracranial aneurysms hemodynamics and predict the effect of different bypass strategies on aneurysm occlusion and ischemic complications [74].

## 5. Conclusions

Treatment of complex anterior circulation aneurysms still represents a challenge for both neurosurgeons and neuroradiologists. The use of revascularization strategies remains a valid option in selected cases where a simpler direct approach, including the use of FD, is not feasible. Our series showed that a patient-specific tailored strategy, based on the specific anatomy and neurophysiology of each individual, including multimodality treatment and a strict collaboration between all the components of a tertiary referral hospital with an expert neurovascular team, is of paramount importance to obtain the best immediate and long-term result for this difficult disease.

## Figures and Tables

**Figure 1 brainsci-12-01339-f001:**
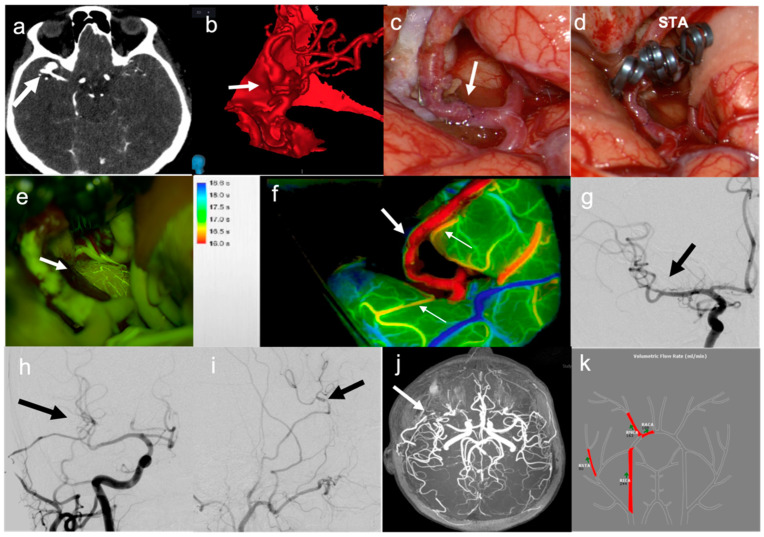
Illustrative case of “Complex” aneurysms with only surgical treatment (bypass and trapping): (**a**) CT-angio showing M2 trifurcation aneurysm (arrow); (**b**) 3D reconstruction of aneurysm (arrow); (**c**) Intraoperative view of STA–MCA bypass (arrow); (**d**) Intraoperative view of aneurysm trapping; (**e**) Intraoperative SF-VA showing the patency of the bypass with an adequate parenchymal perfusion as detectable by the visualization of small cortical vessels; (**f**) FLOW800 analysis showing bypass patency (large arrow) and fronto-temporal branches perfusion (thin arrows); (**g**) Postoperative DSA showing aneurysm exclusion (arrow); (**h**,**i**) Postoperative DSA showing STA–MCA bypass patency in frontal and lateral view (arrow); (**j**) Follow-up Angio-MRI showing bypass patency; (**k**) NOVA MRI showing a good arterial flow.

**Figure 2 brainsci-12-01339-f002:**
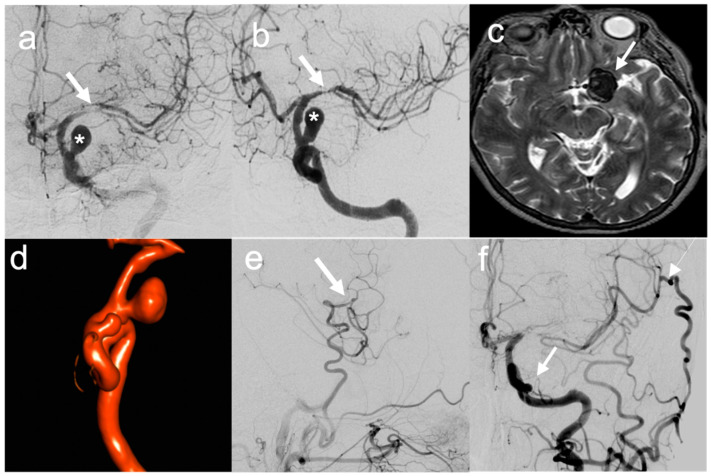
Illustrative case of “Complex” aneurysms with multidisciplinary treatment (bypass for flow preservation due to MCA compression/stenosis and endovascular coiling): (**a**,**b**) Preoperative DSA showing in frontal and lateral view MCA stenosis (arrow) and ICA aneurysm (*); (**c**) Preoperative T2 MRI showing aneurysm partial thrombosis; (**d**) 3D reconstruction of ICA aneurysm and identification of a second ICA aneurysm; (**e**) Postoperative DSA showing STA–MCA bypass patency in lateral view; (**f**) Follow-up DSA showing hypertrophic bypass patency (thin arrow) and aneurysm endovascular occlusion (large arrow).

**Figure 3 brainsci-12-01339-f003:**
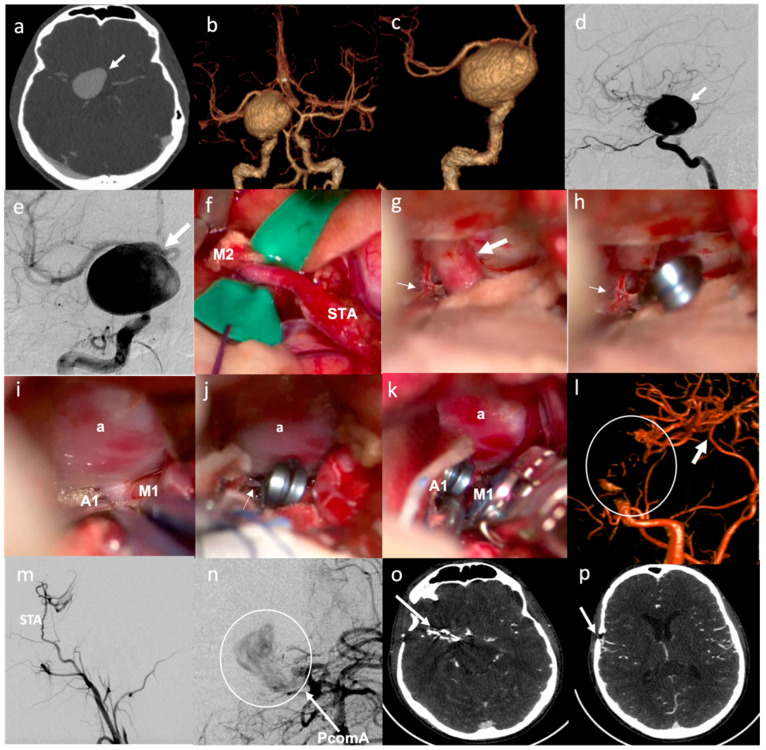
Illustrative case of “Complex” aneurysms with only surgical treatment (bypass and partial trapping because of subsequent aneurysm thrombotic occlusion): (**a**) CT-angio showing giant R ICA aneurysm (arrow); (**b**,**c**) 3D reconstruction of aneurysm (arrow); (**d**,**e**) Preoperative DSA showing R ICA aneurysm (arrow) in sagittal and coronal view; (**f**) Intraoperative view of STA–MCA bypass; (**g**,**h**) Intraoperative pre- and post-surgical clipping view of supraclinoid ICA (large arrow) and ophthalmic artery (thin arrow); (**i**–**k**) Intraoperative view of distal clipping (aneurysm = a); (**l**) Postoperative 3D reconstruction showing aneurysm trapping (circle) and bypass patency (arrow); (**m**,**n**) Postoperative DSA showing STA–MCA bypass patency in lateral view and aneurysm partial trapping with slow perfusion flow from PcommA; (**o**,**p**) Follow-up CT-angio showing bypass patency and aneurysm occlusion.

**Figure 4 brainsci-12-01339-f004:**
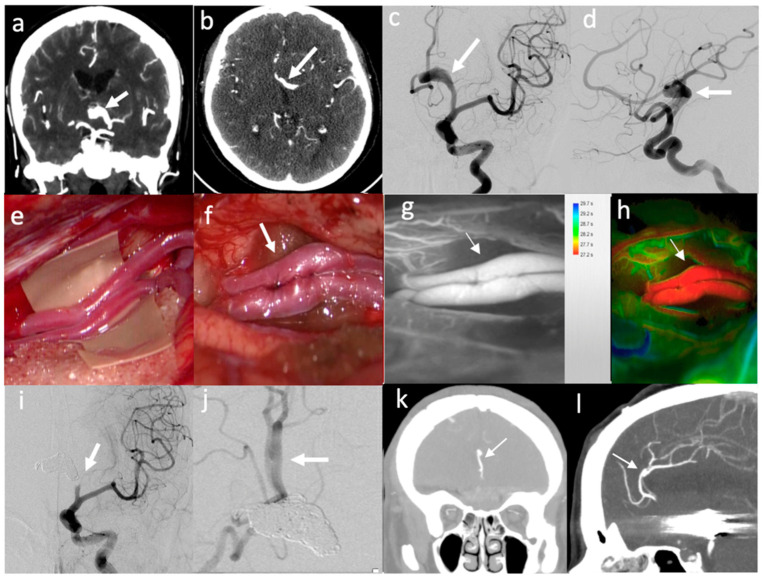
Illustrative case of “Complex” aneurysms with multidisciplinary treatment (bypass for flow preservation and endovascular proximal occlusion): (**a**,**b**) CT-angio showing large fusiform L ACA aneurysm (arrow); (**c**,**d**) Preoperative DSA showing L ACA aneurysm (arrow); (**e**,**f**) Intraoperative view of PerA–PerA latero-lateral bypass (arrow); (**g**,**h**) Intraoperative ICA–VA and FLOW800 analysis showing bypass patency with good perfusion (arrows); (**i**) Postoperative DSA showing endovascular aneurysm trapping with coiling; (**j**) Postoperative DSA showing PerA–PerA bypass patency with flow preservation; (**k**,**l**) Follow-up CT-angio showing bypass patency and aneurysm occlusion.

**Table 1 brainsci-12-01339-t001:** Summary of aneurysm’s characteristics and Intraoperative Data.

		Aneurysm Characteristic				Intraoperative Data				
CASE	Location	Size/Shape/ Rupture Status	Circle of Willis Variations	Intramural Thrombosis	Calcification	Vessels Included	Previous Treatment	Bypass Strategy	Aneurysm Exclusion Strategy	Intraoperative Tool
1	L MCA	32 mm/ Fusiform	\	Partially thrombosed	Wall Calcification	M1	\	STA–MCA Double Barrel	Trapping	ICG-VA/FLOW 800/CEUS
2	L MCA (Distal)	14 mm/ Fusiform	\	Partially thrombosed	\	M4	\	STA–MCA	Distal Clipping	ICG-VA/FLOW 800/CEUS
3	R MCA	35 mm/ Fusiform	Hypoplastic L A1	Partially thrombosed	Wall Calcification	Lenticulostriate	\	STA–MCA Double Barrel	Distal Clipping	ICG-VA/FLOW 800
4	R ICA (intracavenous)	18 mm/ Saccular	Duplicated L PComA	\	\	\	\	EC–IC Radial Graft	Trapping	ICG-VA/FLOW 800
5	L ACA (A1-A2)	40 mm/ Saccular	\	\	Wall Calcification	\	Previous Coiling	PerA–PerA/r STA–r CMA with l STA graft	Endovascular Occlusion	ICG-VA/FLOW 800/CEUS
6	L ICA	15 mm/ Saccular	Absent L PcomA	Partially thrombosed	Wall Calcification	\ (Compression M2 Stenosis)	\	STA–MCA	Endovascular Stenting	ICG-VA/FLOW 800
7	R ICA (intracavernous)	20 mm/ Fusiform	Dysplastic L PcomA, Hypoplastic R A1	\	Wall Calcification	ICA	\	STA–MCADouble Barrel	Endovascular Occlusion	ICG-VA/FLOW 800
8	R MCA	10 mm/ Fusiform	Hypoplastic R A1	\	\	M2	\	STA–MCADouble Barrel	Trapping	ICG-VA/FLOW 800
9	L MCA	12 mm/ Saccular/ ruptured	\	\	\	\	Previous Coiling	STA–MCADouble Barrel	Trapping	ICG-VA/FLOW 800/CEUS
10	R ACA (A1-A2)	5 mm/ Saccular	Unilateral L fetal-type variant with ipsilateral dominant PComA and hypoplastic P1	\	\	A1-A2	Previous Coiling	PerA–PerA	Endovascular Occlusion	ICG-VA/FLOW 800/CEUS
11	L PerA—CMA	16 mm/ Fusiform	\	\	WallCalcification	L CMA	\	PerA–PerA/CMA–CMA	Endovascular Occlusion	ICG-VA/FLOW 800/CEUS, fluorescein
12	L MCA	9 mm/ Saccular/ ruptured	Bilateral hypoplastic PComA	\	\	M2	\	STA–MCADouble Barrel	Trapping	ICG-VA/FLOW 800/CEUS/fluorescein
13	R ICA (intracavernous)	21 mm/Saccular	Bilateral hypoplastic PComA	\	Wall Calcification	ICA	Previous Coiling	STA–MCADouble Barrel	Endovascular Occlusion	ICG-VA/FLOW 800
14	R MCA	45 mm/ Saccular/ ruptured	\	\	\	M2	\	STA–MCA Double Barrel	Trapping	ICG-VA/FLOW 800
15	AcomA	22 mm/Saccular	\	\	\	\	\	PerA–PerA	Endovascular Occlusion	ICG-VA/FLOW 800
16	L ICA	25 mm/Saccular	Absent L PcomA	(R ICA Giant thrombosed Aneurysm)	\	ICA	\	EC–IC Radial Graft/OA–MCA	Endovascular Occlusion	ICG-VA/FLOW 800
17	R MCA	7 mm/Fusiform	\	\	\	M2 (trifurcation)	\	STA–MCA	Trapping	ICG-VA/FLOW 800/CEUS/fluorescein
18	L ACA (A1)	33 mm/Saccular	\	\	Wall Calcification	Perforating artery	\	PerA–PerA	Endovascular Occlusion	ICG-VA/FLOW 800/CEUS
19	L MCA (distal)	20 mm/Saccular	Hypoplastic R A1	Partially thrombosed	Wall Calcification	M3	\	STA–MCA	Clipping (Vessels Ostium Dissection)	ICG-VA/FLOW 800
20	R MCA	14 mm/Saccular	\	Partially thrombosed	\	Temporal M2	\	STA–MCA	Clipping Reconstruction	ICG-VA/FLOW 800
21	R MCA	15 mm/ Fusiform	Unilateral R fetal-type variant with ipsilateral dominant PComA and hypoplastic P1	Partially thrombosed	Wall Calcification	M2		IC–IC (Major Branch)/STA–MCA (Minor Branch)	Trapping	ICG-VA/FLOW 800/CUES
22	R ICA	37 mm/ Fusiform	\	\	\	M1-A1	\	STA–MCA	Partial Trapping and Thrombotic Occlusion	ICG-VA/FLOW 800
23	R ACA	13 mm/Saccular	Bilateral fetal-type variant with dominant PComA and hypoplastic P1	Partially thrombosed	\	PerA–CMA origin	\	PerA–PerA/L A3-Bilateral A4 STA–Graft	Endovascular Flow Diverter	ICG-VA/FLOW 800

**Table 2 brainsci-12-01339-t002:** Summary of preoperative patients’ characteristics.

CASE	Age/ Gender	BMI	Comorbidity	Onset Symptoms	Preoperative mRS	Preoperative KPS
1	67/F	32.77	\	TIA	1	90
2	20/M	25.25	Smoking	Seizure	0	100
3	60/F	19.84	\	Confusion	1	90
4	63/F	27.5	Stage IV renal insufficiency	Pain	0	100
5	59/F	27.34	Hypertension	Seizure	0	100
6	66/F	24.3	\	Anomic aphasia	1	90
7	51/M	31.4	Hypertension	Vertigo	0	100
8	50/F	22.5	Smoking	Incidental finding	1	100
9	35/F	20.6	\	FU in SAH	1	100
10	26/M	29.7	\	FU after endovascular treatment	0	100
11	53/M	26.12	Smoking	Incidental finding	1	90
12	51/M	24.34	\	Headache	1	90
13	74/F	28.4	Hypertension	Diplopia	2	80
14	45/F	21.22	\	SAH	0	90
15	75/F	20.9	Hypertension	Reduction in visual acuity (left eye)	0	90
16	60/F	26.22	Hypertension and smoking	Diplopia (III c.n. palsy), headache	0	80
17	54/F	24.77	\	Headache	1	100
18	66/F	27.41	Hypertension and smoking	Visual field defects	1	80
19	36/F	20.43	\	Incidental finding	0	100
20	62/F	16.42	Hypertension	Headache	0	90
21	66/M	47.75	Diabetes, cardiovascular disease, obesity, smoking	Confusion	0	90
22	19/M	20.9	\	Reduction in visual acuity (right eye)	0	100
23	32/F	33.45	Recent SARS-CoV-2 infection	Incidental finding	1	100

**Table 3 brainsci-12-01339-t003:** Summary of clinical and radiological short- and long-term results.

	Complication		Clinical Data		Radiological Data		
CASE	Intraoperative	Postoperative	KPS at Discharge	Last F-U mRS/KPS	Aneurysm Occlusion at Last F-U	Immediate Bypass Patency	Bypass Patency at Last F-U
1	Aneurysm rupture	Internal capsule stroke (hemiparesis and aphasia)	30	72 months— mRS 2/KPS 80	84 months—Yes	Yes	84 months—Yes
2	\	\	100	84 months— mRS 0/KPS 100	84 months—Yes	Yes	84 months—Yes
3	\	Internal capsule stroke (hemiparesis)	70	36 months— mRS 4/KPS 40	36 months—Yes	Yes	36 months—Yes
4	\	Internal capsule stroke (monoparesis)	80	3 months— mRS 6/KPS 0 ESRD (Death)	2 months—Yes	Yes	2 months—Yes
5	\	\	90	60 months— mRS 0/KPS 100	60 months—Yes	Yes ACA–ACA and STA–ACA	60 months—ACA–ACA Yes/STA–ACA No
6	\	\	100	48 months— mRS 0/KPS 100	36 months—Yes	Yes	48 months—Yes
7	\	\	100	36 months- mRS 0/KPS 100	36 months—Yes	Yes	36 months—Yes
8	\	\	100	48 months— mRS 0/KPS 100	48 months—Yes	Yes	48 months—Yes
9	Extradural hematoma	\	100	48 months— mRS 0/KPS 100	48 months—Yes	Yes	48 months—Yes
10	\	\	100	48 months— mRS 0/KPS 100	48 months—Yes	Yes	48 months—Yes
11	\	\	100	24 months— mRS 0/KPS 100	24 months—Yes	Yes	24 months—Yes
12	\	\	90	9 months—mRS 1/KPS 90	5 months—Yes	Yes	5 months—Yes
13	\	\	80	48 months— mRS 2/KPS 70 (Unrelated, CMT)	48 months—Yes	Yes	48 months—Yes
14	Aneurysm rupture	Internal capsule stroke; evidence of hemorrhagic lesion in the brainstem; deep coma	30	12 months— mRS 6/KPS 0—Death	6 months—Yes	yes	6 months—Yes
15	\	\	90	3 months— mRS 1/KPS 90	3 months—Yes	Yes	3 months—Yes
16	\	Radial artery graft vasospasm with need for angioplasty; left hemispheric hypoperfusion with TIAs	60	24 months— mRS 1/KPS 90	24 months—Yes	Yes EC–IC and OA–MCA	24 months—No EC–IC and OA–MCA
17	\	\	100	24 months— mRS 0/KPS 100	24 months—Yes	Yes	24 months—Yes
18	\	Cerebellar hemorrhage— hydrocephalus treated with VP shunt	40	14 months— mRS 1/KPS 90	14 monthsYes	Yes	14 months—Yes
19	\	\	100	12 months— mRS 0/KPS 100	12 months—Yes	Yes	12 months—Yes
20	\	\	100	5 months— mRS 0/KPS 100	5 months—Yes	Yes	5 months—Yes
21	\	\	100	4 months— mRS 0/KPS 100	4 months—Yes	Yes	4 months—Yes
22	\	\	90	6 months— mRS 0/KPS 100	6 months—Yes	Yes	6 months—Yes
23	Multiple ACA/graft thrombosis and rescue bypass STA to L A3-Bilateral A4	\	40	6 months— mRS 1/KPS 90	4 months—No (FD positioned)	No	4 months—no

## Data Availability

The data presented in this study are available in the present article. Further data on review search or data regarding clinical cases are available upon request to the corresponding author (F.A.).

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
