# Peer review of "The Role of Bypass Surgery for the Management of Complex Intracranial Aneurysms in the Anterior Circulation in the Flow-Diverter Era: A Single-Center Series"

_brainsci, 2022, doi:10.3390/brainsci12101339_

Round 1
Reviewer 1 Report
nothing to comment
Author Response
We thank the reviewer for his/her revision to our study.
Reviewer 2 Report
Please kindly find attached review comments.

Author Response
We thank the reviewer for his/her comments and suggestion to improve our study. Regarding the different points raised by the reviewer,
- The manuscript has been revised by an English native speaker to improve its quality.
- We understand this point and we added the structure of the circle of Willis in Table 1, highlighting cases with variations from the usual configuration.
- Regarding this point, unfortunately we do not have quantitative regional flow measurement performed pre- and post-operatively in every case, thus we could not include these data in the result section of the manuscript. In addition, we have already included the data about bypass patency at long-term follow-up, based on our observation period. In particular, in the new table 3 (we have added a new table 2 based on the request of reviewer 3), data about long-term patency are included in the last column. Nonetheless, following the suggestions from reviewer 2, we have included these points in the new discussion of the manuscript. However, in the article suggested by the reviewer on graft geometry (Seo et al. Semin Thoracic Surg 34:521–532), this factor is evaluated in coronary artery bypasses with venous graft, thus it could not be automatically considered for EC-IC bypass. Furthermore, in our series, most of the cases were operated without using an external graft, but using either IC-IC bypasses or taking advantages of extracranial subcutaneous arteries available in the same area, such as STA or OA.
Reviewer 3 Report
Dear Editor,
The authors evaluated complex IAs with a cerebral bypass. This case can be interesting for the readers of this Journal. The population of this study was not enough for a strong statistical analysis: 11 MCA, 6 ACA, and 6 I CA aneurysms. The discussion section is written very well but Introduction and specifically Result section needs structural corrections. The main concern about this paper is the lack of a strong and clear quantitative parameter to prove this method is more “efficient”. When we want to compare a method or claim that a finding is efficient, we have to pay attention to established and defined statistical policies to ensure the reliability and reproductions of our results. The statistical analysis to prove the correction of results is not enough and it cannot support the hypothesis of this study. I prefer to have the responses to the below comments from the authors.
1. I suggest you add a sentence at the beginning of the Introduction section about the prevalence of IA in the whole world to mention the importance of your work and the necessity of carrying out this study.
2. The Introduction is so brief and you did not provide enough prerequisites to start your main Idea. First, you should give a big picture and general insight to the author and little by little reach the readers to the gap that you want to fill. Computer simulation of the blood hemodynamics and evaluation of rupture risk of the IAs is of great importance to reflect some complexities in IAs [https://doi.org/10.22037/icnj.v3i1.12460], [doi.org/10.1134/S0021894417060025], [doi.org/10.1007/s10143-020-01367-3].
3. “The indications, surgical techniques, … carefully analyzed and presented.” The last sentence of the Introduction section is uncompleted. Can you please write the hypothesis clearly and explain the parameters that you want to use to accept/reject your hypothesis. I can understand them but your hypothesis is not enough clear for the readers.
4. Your IRB approval data is 2012? Can you double-check this date?
5. Can you please add patients’ characteristics in Table 1: age, gender, BMI, etc.
6. I think it would be better if you clarified your intention about “tailored strategy” in the Method section for a better understanding of the readers.
7. I can understand that your definition of “complex” IA limited you to using a large group of patients. However, when we want to compare a method or claim that our result is correct, we have to pay attention to statistical policies to ensure the reliability and reproduction of our results. For example, for your project, I think Kaplan-Meier analysis is necessary. In addition, paying attention to the confidence interval, coefficient of variations and other related statistics is also important.
8. I think “3.1. Patients characteristics” is not a result and this is a part of the Method section. In addition, I think some parts of “3.2. Bypass strategies” is also a part of the Method, not the result. Other subsections of the Result section are also wired to me. You wrote the result section similar to a case report paper. I think most parts of the Result section are wordy or are Method. Because you did not evaluate any specific parameters like morphological parameters (except for diameter) for evaluation or comparison with other types of IAs or other surgery methods. Hence, you did not have any specific data and parameters to report in the Result section. I know the value of your work and I respect your attempt but honestly, this is not the Result. You should report changes in the values of some parameters or you did not compare unique parameters in different surgery methods or types of IAs. How did you understand this method was more efficient? What were your indicators to reject or accept your hypothesis in the Result section.
9. “The use of bypass techniques represents, in selected cases, a valid, safe and effective therapeutic option in the management of complex anterior circulation an eurysms.” This is a very big claim. Your results cannot support this big claim. “valid”, “safe” and “effective” are very sensitive words. For example, when you want to use “effective”, you have to compare some methods and procedures and you have to pass defined and established statistical procedures to have this permission to use this word. I think everything in this paper is qualitative but your conclusion has a great quantitative burden.
10. The discussion section was well-designed and at the end of the Discussion, you can make a separated sub-section to describe the limitations and future works separately.
Author Response
We thank the reviewer for his/her suggestions and comments about our study. We are fully aware of its limitation regarding the number of cases and the lack of a control group. However, although limited, our series is representative of a selected population of complex aneurysms in which the use of bypass represent a possible therapeutic option. Furthermore, we did not want to claim that our strategy was superior to other available treatments, but we wanted to show, on the opposite, that this was a viable alternative when other therapeutic options including flow diverters, in our hands, were considered not feasible. That is the reason why we could not have any control group in such a study.
Regarding the different points raised by the reviewer:
- Following the reviewer’s suggestion, a sentence has been added about the prevalence of IA in the whole population and the importance of carrying out our study.
- We agree with the authors that blood hemodynamics could have a possible impact on the rupture risk of unruptured IA. However, our focus was on complex aneurysms, in which a simple endovascular or surgical approach, including flow diverter, was considered not feasible. We did not want to address the specific risk of rupture of complex aneurysm or unruptured IA in general. We tried to report on the use of different bypass strategies to preserve distal flow, together with therapeutic options to occlude the aneurysm, when other options, in our hands, were considered not feasible. We modified the introduction to make our background and aims clearer, including also recent data coming from the ESO guidelines on management of UIAs, published in September 2022.
- The sentence has been restructured and included in a modified version of our introduction, based on the other points raised in the revision. However, our study is a retrospective analysis of the results of a case series, and there is no control group to evaluate whether our approach is better or not, as compared to another. Thus, the is no hypothesis to accept or reject. We tried to make this point clearer in the new version of the introduction, including the last sentence.
- Yes this is correct, as this is the date when our prospective surgical database on complications, from which the data from this study were retrospectively retrieved, has been approved. We better specify this issue in the material and methods, as follow: “The Ethical Committee of the Foundation IRCCS Istituto Neurologico Carlo Besta approved the prospective surgical database of the Neurosurgical Department (4/4/2012) that was used to perform this retrospective analysis.”
- We agree with the reviewer and we included a new table (Table 2) to add all the requested data, including also pre-operative conditions
- A specific sentence in the introduction was already present to specify that a tailored approach should be considered in complex aneurysm, based on specific patient’s anatomy and aneurysm characteristics. We have also modified the discussion to better specify the idea of tailored strategy of bypass (“By mastering the three vascular microanastomosis (i.e. end-to-end, end-to-side and side-to-side) [54] it is possible to hypothesize different creative forms of revascularization, including EC-IC or IC-IC bypasses, that could be adapted and tailored to the specific patient’s anatomy and to the precise need for flow revascularization [21,25]”). We have also modified the first sentence of paragraph 2.2 and 3.3 as follow: “The bypass modality was tailored on the aneurysm characteristics and on the brain territory to be supplied (Table 1)” and “The actual strategy of aneurysms’ occlusion was also tailored to the location and shape of the aneurysm, and on involvement of perforators (Table 1)”
- As already stated in previous responses to other points, our study is a retrospective case series analysis, based on inclusion and exclusion criteria decided a priori. Thus, there is no confidence interval, Kaplan Meier comparative analysis or other statistical analysis that can be performed as no comparison has been made with a control group; furthermore, we have no way to artificially enlarge our surgical group as this is an honest analysis of our surgical series of bypass in the last years, as it really is. Based on the number of the case series and its heterogeneity, we could not perform any subgroup analysis to evaluate the impact of different strategies on outcome. We have addressed this limitation in a specific paragraph of our discussion.
- We do not fully agree with this point raised by the reviewer. In the material and methods we reported the inclusion criteria and how we retrospectively analyzed our series, and also which were the criteria for choosing the type of bypass and the strategy of aneurysm occlusion in our series of complex aneurysms in the anterior circulation. In the result section, we defined which were the patients included in the study, and which were the bypass techniques and how the aneurysm was occluded; in addition, we reported immediate and long-term clinical and neuroradiological results and complications. These, in our mind, are actual results of a retrospective case series. We do understand that some paragraph of the results may sound similar to a case report, but as each complex aneurysm could have a specific type of configuration and treatment, we believed that it was important for the reader to have information about all the bypass modalities and type of aneurysm treatment performed in our series. Furthermore, due to the low number of cases, the absence of control group, and the fact that in most of the cases a good outcome was obtained at long-term follow-up, no subgroup analysis evaluating some factors possibly related to outcome was performed. However, according to the reviewer’s suggestion, we tried to simplify all the paragraphs to make the manuscript less wordy, and we addressed the limitation in the revised version of the discussion.
- The sentence the reviewer was referring to was present in the abstract, and it was modified as follow: “The use of bypass techniques represents a valid therapeutic option in the management of complex anterior circulation aneurysms, when a simpler direct approach, including the use of FD, is considered not feasible”
- We agree with the reviewer and we included a separate sub-section to describe the limitations of the study and future directions
Round 2
Reviewer 2 Report
Thanks for the update. The majority of my earlier comments have been addressed. Considering the hemodynamic effect of the Circle of Willis on the aneurysm and the graft, it would be interesting to explore the usage of computational models of CoW in optimizing the surgical planning in future studies.
Reviewer 3 Report
Accept